# Compression of Hyperspectral Scenes through Integer-to-Integer Spectral Graph Transforms †

**Dion Eustathios Olivier Tzamarias** *, **Kevin Chow**, **Ian Blanes** and **Joan Serra-Sagristà** 

Department of Information and Communications Engineering, Universitat Autònoma de Barcelona, Campus UAB, 08193 Cerdanyola del Vallès, Barcelona, Spain; kevin.chow@uab.cat (K.C.); ian.blanes@uab.cat (I.B.); joan.serra@uab.cat (J.S.-S.)

* Correspondence: dion.tzamarias@uab.cat
† This paper is an extended version of our paper published in in the Proceedings of the 6th ESA/CNES International Workshop on On-Board Payload Data Compression (OBPDC), Matera, Italy, 20–21 September 2018, titled *Compression Of hyperspectral images with graph wavelets*.

**Abstract:** Hyperspectral images are depictions of scenes represented across many bands of the electromagnetic spectrum. The large size of these images as well as their unique structure requires the need for specialized data compression algorithms. The redundancies found between consecutive spectral components and within components themselves favor algorithms that exploit their particular structure. One novel technique with applications to hyperspectral compression is the use of spectral graph filterbanks such as the GraphBior transform, that leads to competitive results. Such existing graph based filterbank transforms do not yield integer coefficients, making them appropriate only for lossy image compression schemes. We propose here two integer-to-integer transforms that are used in the biorthogonal graph filterbanks for the purpose of the lossless compression of hyperspectral scenes. Firstly, by applying a Triangular Elementary Rectangular Matrix decomposition on GraphBior filters and secondly by adding rounding operations to the spectral graph lifting filters. We examine the merit of our contribution by testing its performance as a spatial transform on a corpus of hyperspectral images; and share our findings through a report and analysis of our results.

**Keywords:** hyperspectral image coding; graph filterbanks; integer-to-integer transforms; graph signal processing

## 1. Introduction

Hyperspectral images are representations of scenes across many bands of the electromagnetic spectrum (otherwise called spectral components). These images have been used to classify areas of a landscape [1], identify clouds [2] or seeds located on the Earth's surface [3] and even forecast geological events such as landslides [4]. Due to their large size, these images require the use of efficient compression techniques in the form of specialized image transforms. The above mentioned transforms can often be computationally costly but one could strive towards real-time compression of remotely-sensed hyperspectral images, when processed accordingly through on-ground, high-performance parallel computing facilities.

Transform-based lossy compression techniques used on hyperspectral images are commonly related to one of two families. The first group of algorithms are based on the Discrete Wavelet transform (DWT) [5–8]. On one hand the advantage of these transforms manifests through their low computational complexity, though they are not adaptive to the input signal. On the other hand, algorithms such as those in References [9–12] are based on the Karhunen-Loeve transform (KLT), which is adaptive at the expense of a high computational complexity. Meanwhile the techniques used for the

lossless compression of hyperspectral images are generally based on a predictive coding model [13,14] though lossy predictive techniques [15] have the benefit of a lower computational complexity. Some lossless predictive techniques [16,17] are implemented in the CCSDS-123.0-B-2 [18] standard that also supports near-lossless compression and whose parameter tuning is explored in Reference [19].

This paper exploits a recently emerged family of wavelet transforms which are based on graph signal processing, a field in which data structures are represented as signals lying on the nodes of graphs while weighted graph edges denote the degree of similarity between nodes [20,21]. Graph wavelet transforms are derived from the graph structures, which in turn, one can also construct from images. Applying such transforms on hyperspectral images has been shown to lead to competitive compression results [22].

The strength of graph wavelet transforms in the field of image compression stems from the Graph Fourier Transform (GFT), an equivalent to the classical Fourier transform that is formed by the eigenvectors of the graph Laplacian, a matrix constructed from the values of the weighted edges of the graph. Just like the KLT, the GFT is an adaptive transform but instead of following a statistical approach, it attempts to encode image structures that are embedded in the graph edges. Thus, an important attribute of the GFT is related to its flexibility, since one can decide on the degree of accuracy with which image structures are represented on the graph [23]. It has also been shown [24] that not only does the GFT approximate the KLT for a piece-wise first-order autoregressive process but also that the GFT optimally decorrelates images following a Gauss-Markov random field model [25]. Consequently, graph wavelet transforms combine the GFT with a multiresolution analysis, resulting in a powerful tool for image compression.

A brief review of graph wavelet transforms follows. One can organize these transforms into two general groups—vertex domain and spectral domain designs. The former are based on spatial features of the graph such as the degree of connectivity between nodes. These vertex domain designs, though, lack spectral localization. While the energy of the resulting transformed signal is not concentrated around central graph frequencies, the vertex domain designs are described by a perfect localization on the vertex domain, meaning that one can define the number of nodes that will update the value of a vertex after the transform. One specific vertex domain design is the lifting-based graph wavelets [26] that use distinct groups of nodes to compute the update and detail wavelet coefficients.

Transforms that belong in the latter category of spectral designs exploit characteristics of the graph spectrum (not to be confused with the spectrum of hyperspectral images) in the form of the eigenvectors and the eigenvalues of the graph. One key feature of these designs is good spectral as well as vertex domain localization. One of the first spectral graph wavelet transforms are the diffusion wavelets [27] as well as the spectral graph wavelets [28]. While these spectral graph transforms are over-complete (the number of wavelet coefficients surpass the number of signal samples), this problem is solved by the two-channel graph wavelet filterbanks [29] that uses quadrature mirror filters (QMF) on bipartite graphs. These filters are not compactly supported and produce transforms that are not well localized on the vertex domain. The next iteration of such transforms are the biorthogonal graph wavelet filterbanks (introducing the GraphBior transform) [30] which are compactly supported as well as critically sampled. Since then several improvements and variations of the biorthogonal graph filterbanks have been proposed by including spectral sampling [31] or by introducing the M-channel graph wavelet filterbanks that can be implemented on large sparse graphs [32]. One of these variations uses polyphase transform matrices [33], proposing graph lifting structures [34] in the spectral domain for biorthogonal graph filterbanks.

One major issue of the filters developed for biorthogonal graph filterbanks, regarding image compression, is that they are only suited for lossy compression schemes. This drawback originates from the fact that the graph wavelet coefficients arising from these transforms are not integer. Thus a necessary quantization step is required rendering such compression schemes lossy. Furthermore, the use of spectral graph transforms for the compression of hyperspectral images spawns further

difficulties—a huge amount of side information is required, due to the necessity to transmit the graph structure to the decoder.

In this paper, we provide transforms that allow us to construct a lossy-to-lossless compression scheme for hyperspectral images, using biorthogonal graph filterbanks. We developed two integer graph-transforms, both suitable for the biorthogonal graph filterbanks.

Our first approach is to modify the filtering process of GraphBior so that the resulting analysis wavelet coefficients are integer. We solve this problem by computing the Triangular Elementary Rectangular Matrix (TERM) [35] decomposition of the spectral GraphBior filter. At first glance this solutions seems promising but due to the high complexity of the TERM factorization process and the large size of the GraphBior filters, one could argue otherwise. Thus we partition each GraphBior filter in tiles and compute the TERM factorization of each tile in parallel. This process dramatically decreases the time complexity of the TERM factorization of GraphBior.

Our second integer spectral graph transform is achieved by introducing rounding operations within the spectral graph lifting structures proposed in Reference [34]. Hence, we transform it into an integer-to-integer graph wavelet transform.

Using our proposed transforms mentioned above, we design a lossy-to-lossless extension to the scheme developed for the lossy compression of hyperspectral images through GraphBior in Reference [22]. The compression scheme published in Reference [22] tackles the issue of transmitting the graph structure to the decoder by assembling consecutive hyperspectral components into packets called band groups.

In this paper we evaluate and analyze the performance of our lossy-to-lossless compression scheme on a variety of hyperspectral images. We then explore the use of our graph transforms for hyperspectral images. Additional, we explore the effect that different parameters have on our compression scheme. Such parameters are the size of the tiles as well as the size of band groups. Experimental results are provided comparing the performance of the proposed techniques for several images from the CCSDS MHDC corpus of hyperspectral images [36].

This paper is structured as follows. First, in Section 2, we present a brief overview of graph signal processing and biorthogonal graph filterbanks. Then, the derivation of the TERM decomposition of the GraphBior transform, the introduction of the integer spectral graph lifting transform, as well as the adopted coding strategy are discussed in Section 3. In Section 4 we detail the setting of our experiments and showcase our results. Our conclusions are stated in Section 5.

## 2. Graphs and Biorthogonal Graph Wavelets

This section provides a brief introduction on graph signal processing and GraphBior. A graph $G = (V, E)$ is composed by a set of nodes $V$ that are linked to each other by a set of edges $E$. There is a large variety of graphs, each with its own special properties. One relevant example is the bipartite graph, whose nodes can be arranged in 2 subsets such that there are no edges connecting nodes of the same subset. In other words, one needs only two colors in order to color the nodes of the graph in a way that no two nodes of the same color are connected through an edge. Two examples of bipartite graphs can be seen in Figure 1b,c.

The edges of a graph can be weighted, in a way that nodes can be connected strongly or weakly by a non discrete measure. The adjacency matrix $A \in \mathbb{R}^{|V| \times |V|}$, of a graph, is the symmetric matrix that describes the strength of all the possible connections between all nodes of a graph. We establish matrix $A$ in such a way that any of its elements located in row $i$ and column $j$ is a real number $a_{ij} \in [0, 1]$ representing the weight of the edge between node $i$ and $j$. The stronger the connection between two nodes, the higher the value of the weight, with the value 1 describing an edge of the highest strength. Conversely, weaker connections are represented by lower adjacency values such that a weight value equal to 0 represents a non existing edge between nodes. Using the Adjacency matrix one can compute the normalized Laplacian matrix that is defined as $\widetilde{L} = I - D^{-1/2}AD^{-1/2}$, where $D$ is the diagonal degree matrix whose $i$th element is equal to the sum of the elements of $A$ situated on its $i$th row.

In graph signal processing the normalized Laplacian matrix is of vital importance since it embeds within its structure the spectral information of the graph. Due to its symmetric nature, the eigenvectors of $\widetilde{L}$ create a complete orthogonal basis that can describe any vector in $\mathbb{R}^{|V|}$ as a linear combination of its basis. Additionally the eigenvalues of $\widetilde{L}$ are known as the spectrum of the graph and portray a notion of frequency.

We can also assign a real value to each node of the graph such that node *i* has a value of $f_i$. A graph signal $f \in \mathbb{R}^{|V|}$ is expressed mathematically as a vector and represents the collective values of the nodes. A graph signal of low frequency is expected to vary slowly, meaning that strongly connected nodes will support very similar graph signal values, while a high frequency signal is expected to assign very dissimilar values to strongly connected nodes. In other words a low frequency signal reflects the natural connectivity characteristics of the nodes of the graph whereas a high frequency signal goes against the connectivities imposed by the values of the graph edges.

The GraphBior transform, just as classical wavelet filterbanks, filters in the analysis step a signal into low pass and high pass signals that are later downsampled in order to decrease the number of graph wavelet coefficients. In order to guarantee reversibility of the transform, GraphBior exploits a particular characteristic of the spectrum of bipartite graphs (called spectral folding) [30]. Thus an arbitrary graph first needs to be decomposed into a series of bipartite subgraphs. These subgraphs share the same set of nodes as the original graph but their sets of edges do not intersect. The GraphBior filterbanks then makes use of these graphs by creating low and high pass filters.

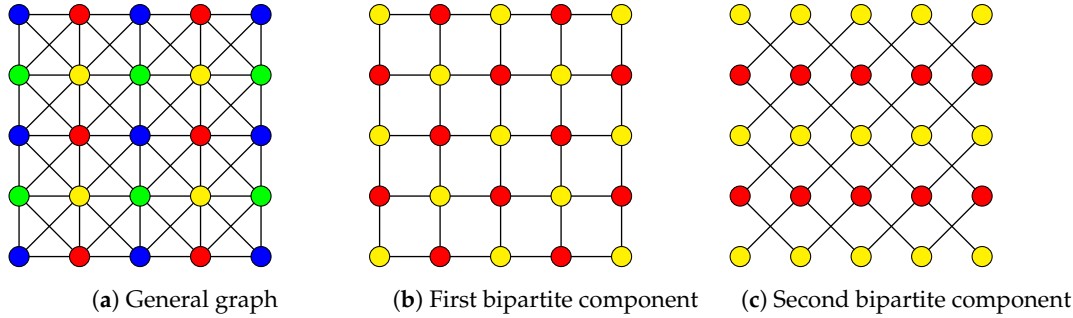

(**a**) General graph      (**b**) First bipartite component      (**c**) Second bipartite component

**Figure 1.** A graph (**a**) is decomposed into bipartite components (**b**) and (**c**). Node colors are the result of the graph coloring process, required for the graph bipartition.

## 3. Lossy-to-Lossless Graph Wavelet Filterbanks

Our work in this paper strives towards a lossy-to-lossless extension of Reference [22] for the compression of hyperspectral images using the GraphBior filterbanks. In this following section we go through the compression scheme that is employed, which follows the one introduced in Reference [22]. Then we propose an integer-to-integer version of the GraphBior filters, by applying a TERM decomposition, as well as tiling to decrease the time complexity of the TERM factorization. Additionally, we propose a second integer-to-integer graph transform suited for the biorthogonal graph filterbanks by modifying the spectral graph lifting structures in Reference [34].

### 3.1. Compression Scheme

Our compression scheme consists of 3 modules—one whose purpose is to calculate the biorthogonal graph filterbank transform, an encoder and a decoder.

The first module is where a single hyperspectral component is used for the construction of a biorthogonal graph filterbank transform and follows the scheme introduced in Reference [22]. The component is first mapped into a graph, then decomposed into a series of bipartite graphs which will finally be used to create the graph transform. We design a graph *G* by representing the pixels of a component as graph nodes and connect neighboring nodes with vertical, horizontal and diagonal edges resulting in a 8-regular graph just as the one shown in Figure 1a. Edges between two connected

nodes are strong if the luminance values of these two pixels are similar and low when they are not. Specifically a graph edge $a_{ij}$ is calculated through the Gaussian kernel $a_{ij} = e^{-\frac{(f_i - f_j)^2}{\sigma}}$ where $f_i$ and $f_j$ are the values of pixels $i$ and $j$ and $\sigma$ is a scaling factor. Once the graph $G = (V, E)$ has been constructed, it is decomposed into a series of $n$ bipartite subgraphs $B_i = (V_i, E_i)$ with $i = 1, \ldots, n$. The bipartition should be done in a way where each bipartite subgraph has the same vertex set as the original graph, whereas the edges of $E$ are distributed among the subgraphs $B_i$ in such a way that no single edge of the original graph is present in more than one bipartite subgraphs. Thus, the union of all the sets of edges from all the bipartite graphs is equal to the set of edges of the original graph. In other words, $V_i = V$, $\cup_i E_i = E$ and $E_i \cap E_j = \varnothing$ for $i \neq j$. To decompose a graph into a series of bipartite subgraphs, we utilize Harary's decomposition [37]. Other decomposition techniques are found in References [38–40]. In the case of the 8-regular graph, shown in Figure 1a, the bipartition resulting from Harary's algorithm is very intuitive and is not computationally intensive. By discarding all diagonal edges from the graph in Figure 1a, we construct the first bipartite graph, shown in Figure 1b and by discarding all horizontal and vertical edges from Figure 1a, we construct the second bipartite graph, shown in Figure 1c. Hence, an 8-regular graph $G$, constructed out of a single component, is decomposed into two bipartite graphs that are utilized for the computation of a biorthogonal graph filterbank transform, just as shown in Figure 2.

It is necessary, for the reversibility of the transform, that the encoder and the decoder have both access to the same graph structure. This ensures that both modules construct the analysis and synthesis filters out of the same graph. The first module is designed in a way that circumvents the transmission of the graph structure to the decoder without the need of side information. This is done by only using decoded components for the calculation of the graph wavelet transforms. Specifically, components are bundled into packets of consecutive components called band groups. Each band group should preferably consist of the same number, $\omega$, of components. Once the decoder has decompressed a band group we extract the last of its components. Using that component we create the graph $G$ and thus the graph wavelet transform. This process can be executed in the decoder and the encoder simultaneously. Once the graph wavelet transform has been learned, it is then applied to the next band group, in the analysis step within the encoder as well as in the synthesis step at the decoder. A schematic representing the computation and usage of the graph filterbank transform is depicted in Figure 3a.

The next module is the encoder, whose first operation upon receiving an image is to partition all but the first component into band groups. The first component is then compressed in a lossless manner by a spatial DWT and transmitted to the decoder. Thus the first component can be used to create the biorthogonal graph filterbank transform for the first band group. Remaining components are partitioned in band groups and encoded sequentially. The next step involves an optional RKLT or DWT transform across the spectral direction of the band group. This step is followed by the application of a spatial graph wavelet transform and the resulting graph wavelet coefficients are then quantized and coded by a plain entropy encoder. Accordingly, the module of the decoder is composed by an entropy decoder, a dequantizer, the synthesis process of the biorthogonal graph filterbank and the optional inverse spectral transform. The encoder and decoder modules are depicted in Figure 3b. By introducing our integer-to-integer filters for the biorthogonal graph filterbank we obtain a lossy-to-lossless compression scheme using graph wavelets.

It is important to note that this manuscripts aims to study graph wavelet transforms when applied spatially to hyperspectral images. For that reason, we have followed a minimalist approach to designing our overall compression scheme. By avoiding complex quantization or entropy encoding stages, with complex overall interactions, we focus our efforts to the proposed graph transforms and to measure their performance with less interference. While a more complex integration of post-transform encoder parts would be possible, due to the adaptive nature of the biorthogonal graph filterbanks, it is expected that these transforms perform competitively, regardless of the probabilistic characteristics of the pre-transform data. For theoretical result on transform performance, readers can see Reference [25],

where it is proven that the GFT optimally decorrelates images following a Gauss-Markov random field model.

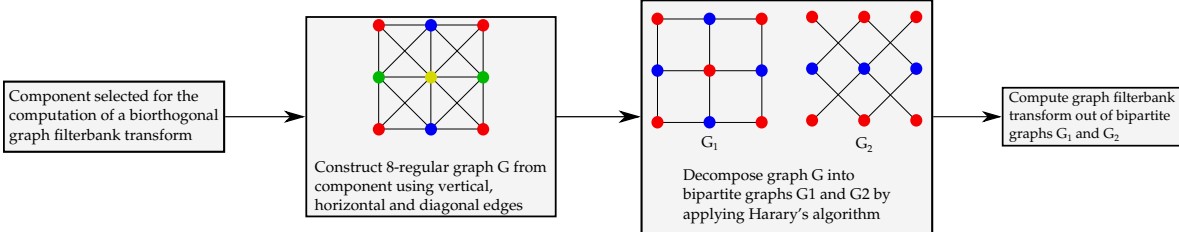

**Figure 2.** Construction of the bipartite graphs out of a selected component leading to the calculation of a graph filterbank transform.

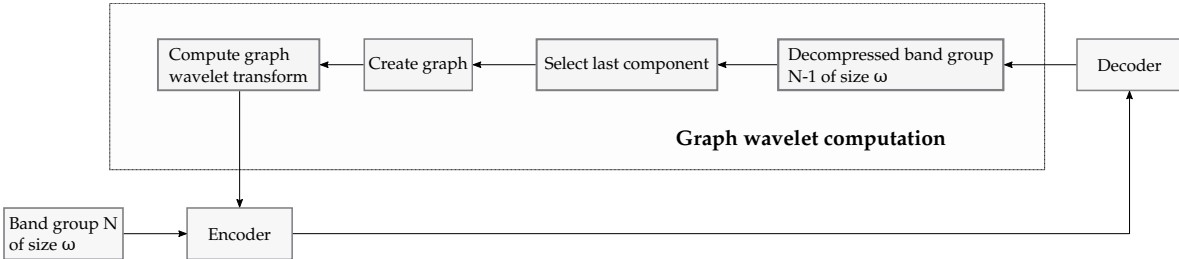

(**a**) Computation of graph wavelet filterbank transform

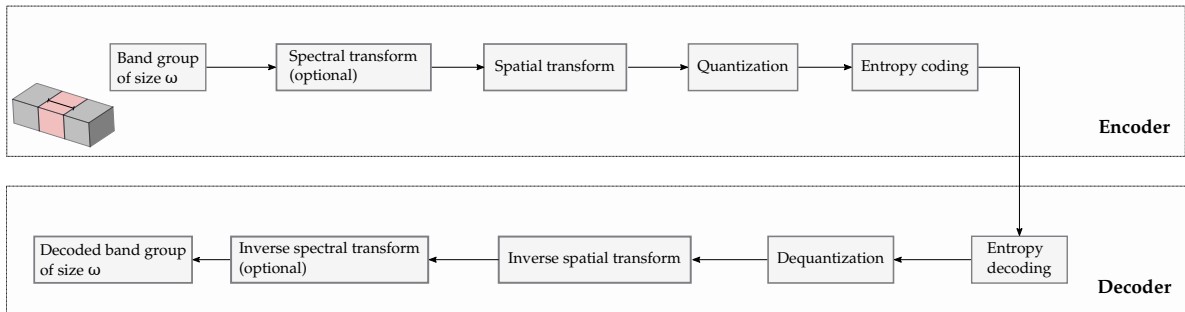

(**b**) Compression scheme

**Figure 3.** Schematics of the compression scheme used.

### 3.2. TERM factorization of GraphBior

In order for the proposed graph compression scheme, based on the biorthogonal graph filterbanks, to develop a lossless behavior we require an appropriate transform. Specifically, an integer-to-integer spectral graph wavelet transform. One of our solutions is to apply the TERM factorization process and the computational scheme introduced in Reference [35] on the GraphBior transform matrix.

However, one cannot decompose any arbitrary square transform matrix through TERM. The necessary condition in order to apply such a factorization dictates that the determinant of the transform should be strictly equal to 1. By design the GraphBior transform is reversible, which means that the determinant of the GraphBior analysis filter $T_a \in \mathbb{R}^{|V| \times |V|}$ is equal to an arbitrary non zero real number $d$.

We scale the elements of $T_a$ by $d^{-1/|V|}$, without the loss of graph spectral information, to force its determinant to 1. Therefore we create a new matrix $T'_a = T_a \cdot d^{-1/|V|}$ whose determinant is equal to 1 and can follow the decomposition and computational scheme of Reference [35]. Applying the TERM factorization to the transform $T'_a$ and by following the computation scheme introduced in Reference [35] we manage to create an integer-to-integer variation of the GraphBior transform. We shall abbreviate our proposed TERM GraphBior transform as IGB (integer GraphBior).

This factorization procedure raises the concern for certain drawbacks, specifically if we consider the large size of the GraphBior transform matrix. These square nonsingular matrices have as many rows (or columns) as the total number of pixels in each hyperspectral component. Due to its cubical computational complexity, the factorization of a GraphBior transform for an entire component, through TERM, is computationally intensive. As a first measure we use low order polynomials to compute the GraphBior filters. For that reason we use the GraphBior(1,1) rather than the better performing GraphBior(5,5) transform, since the former produces a sparser transform matrix. Additionally we split each component that is used to create the transform into smaller square tiles in an effort to accelerate the factorization process.

### 3.3. Tiling

In order to alleviate the time complexity from the TERM factorization, we developed a divide and conquer method. We initially split the component that is used to derive the graph wavelet transform into multiple small square tiles. From each tile, we compute its corresponding GraphBior transform to which we apply the TERM factorization. For the selection of a suitable tile size, in this subsection we study the effect on the performance of our compression scheme with respect to the tile size. Therefore we have experimented with IGB on our compression scheme, with no spectral transforms and using 3 different tile sizes (8 by 8, 16 by 16 and 32 by 32 square tile sizes). We tested the performance of IGB on the hyperspectral images, listed in Table 1, that contain several hyperspectral scenes available at the CCSDS website [36]. In Table 2 we display the entropy, in bits per pixels per component (bpppc), at which IGB becomes lossless, while using $\omega = 2$. We observe that the entropy where the IGB transform becomes lossless decreases as the tile size increases. This first observation encourages us to use a larger tiles for the lossless case. The results from Figure 4 lead to the same conclusions when testing the lossy performance of IGB. Specifically, in Figure 4 we display the performances of IGB when using tiles of 8 by 8 and 16 by 16 pixels, relative to using larger tiles of 32 by 32 pixels. Larger tile sizes are not included in our experiments as we are limited by the high computational complexity of the TERM decomposition. In Figure 5 we have repeated the same experiment on other hyperspectral images. Regarding the two AVIRIS images, due to their much larger spectral size, we have not experimented with the largest tile option as we have done with the Landsat image. The experimental conclusions also agree with our intuitive interpretation of the graph biorthogonal filterbanks since when using larger tiles, the graph can exploit spatial redundancies on a bigger area of each component. This is because the graph edges that connect adjacent nodes that belong to different tiles are not utilized. Since the number of discarded edges rises as the tiles becomes smaller so does the performance of IGB deteriorate. Therefore it is beneficial to keep the sizes of the tiles as large as possible but small enough to allow a fast TERM factorization. Luckily this solution can be parallelized since we could process every tile of every component of the same group at the same time.

**Table 1.** The hyperspectral images with their dimensions used in our experiments.

| Name | Instrument | Calibrated | Across-Track | Along-Track | Spectral Dimension |
|---|---|---|---|---|---|
| Yellowstone sc. 0 cal. | AVIRIS | Yes | 512 | 512 | 224 |
| Yellowstone sc. 0 raw | AVIRIS | No | 512 | 512 | 224 |
| Lake Monona | Hyperion | Yes | 512 | 256 | 242 |
| Mt. St. Helens | Hyperion | Yes | 512 | 256 | 242 |
| Agriculture | Landsat | No | 512 | 512 | 6 |

**Table 2.** Rates at which integer GraphBior (IGB) achieves a lossless compression using $\omega = 2$ for various tile sizes and multiple images. Units are in bpppc.

| Image | IGB | | |
|---|---|---|---|
| | **Tiles of 8 by 8** | **Tiles of 16 by 16** | **Tiles of 32 by 32** |
| Yellowstone sc. 0 cal. | 7.14 | 6.95 | - |
| Yellowstone sc. 0 raw | 9.15 | 9.02 | - |
| Lake Monona | 6.40 | 6.31 | 6.26 |
| Mt. St. Helens | 6.78 | 6.68 | 6.63 |
| Agriculture | 4.39 | 4.27 | 4.21 |

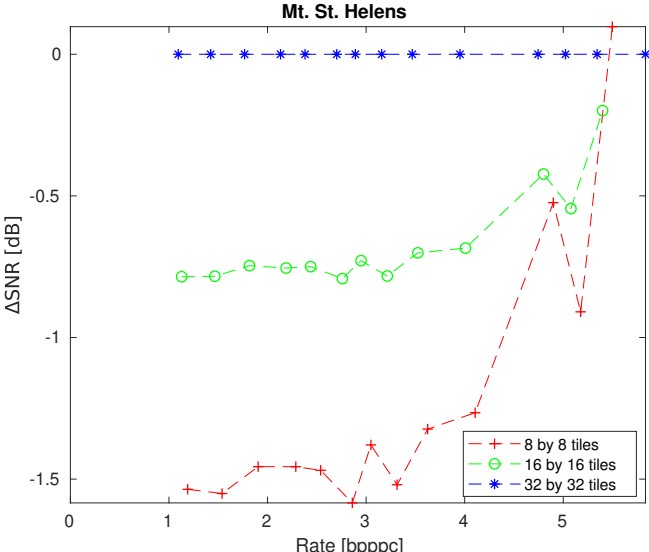

**Figure 4.** Relative rate-distortion plots for IGB, using $\omega = 2$, when varying the tile size. Results are relative to those of the largest tile size.

### 3.4. Integer-to-Integer Spectral Graph Lifting

The main disadvantage of the IGB arises from the computational complexity of the TERM factorization step. As we have seen in the previous section although the tiling strategy reduces the time complexity for the TERM factorization, the performance of our compression scheme decreases in relation to the size of the tiles. For that reason we introduce a second integer-to-integer graph filterbank transform using the spectral graph lifting filters [34].

These filters are arranged into Type 1 and 2 polyphase transform matrices (PTM) which have an upper and lower triangular structure respectively. The final transform matrix is calculated by multiplying alternatively Type 1 PTMs with Type 2 PTMs. One can notice that both types of PTMs are unit triangular rectangular matrices, which means that by following the TERM filtering process we can modify the spectral graph lifting filters into integer-to-integer transforms. Due to their particular structure, by introducing rounding operations after applying each PTM matrix multiplication, the transform becomes integer and reversible. We experiment with the integer-to-integer spectral graph lifting using the quad kernel (ISGL$_Q$) and the integer-to-integer spectral graph lifting using the dual kernel (ISGL$_D$) designs [41].

It should be noted that the computational complexity of this transform is comparable to the one of the GraphBior transform and can be easily computed from an entire component without the need of tiling operations. Specifically the computational complexity of the costlier ISGL$_Q$ is comparable to the one of GraphBior(5,5) since the former requires a computation of a 10th and 13th degree matrix polynomial, whereas the latter one of 10th and one of 11th. This means that by implementing the filtering process with Chebyshev polynomials [28] one can use the same order $M$ to approximate

similarly well the GraphBior(5,5) as well as the ISGL$_Q$ filters resulting in a computational complexity of $O(M|E|)$ per filter, where $E$ is the set of edges of the graph. By performing the appropriate modifications, our proposed ISGL$_Q$ compresses and decompresses the calibrated AVIRIS image from Table 1 in 4.5 min, whereas the IGB required more than 1 h to successfully perform the same action on a computer using an Intel Core i7-7700HQ CPU and 16GB of RAM.

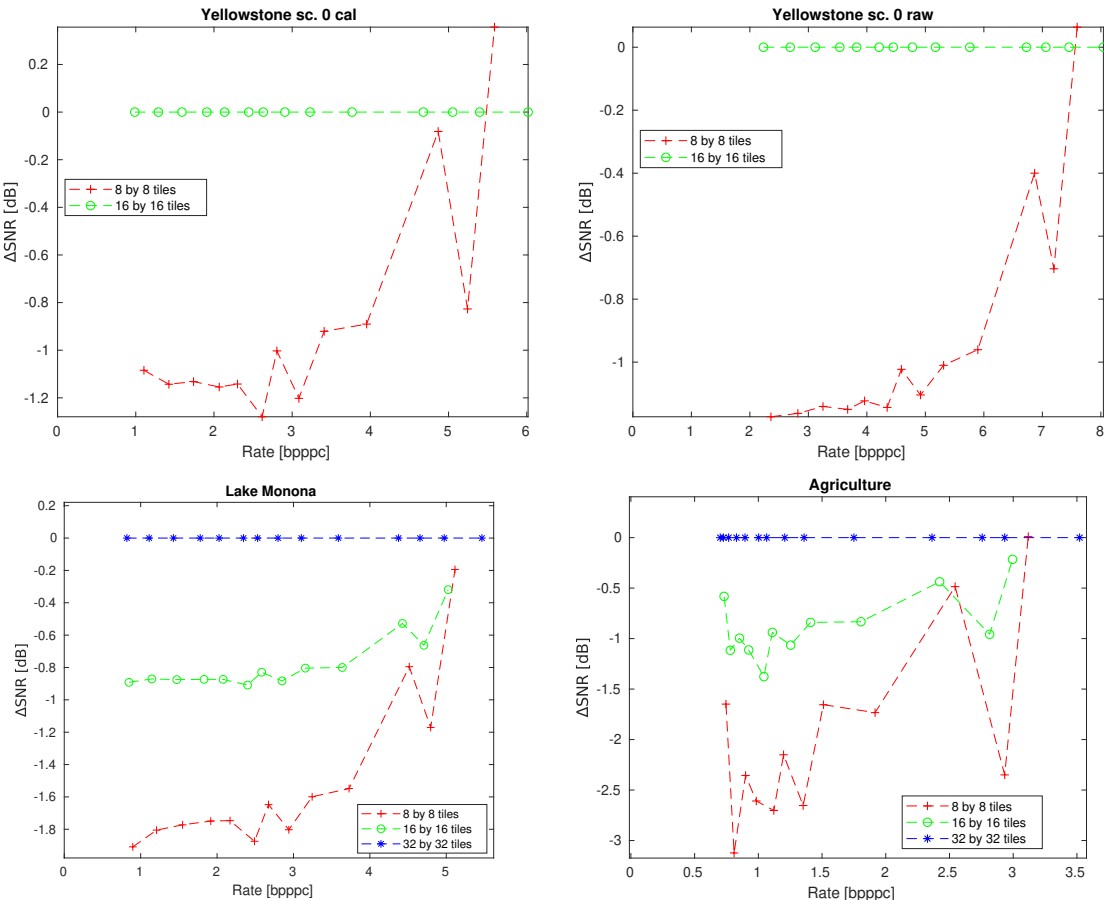

**Figure 5.** Relative rate-distortion plots for IGB, using $\omega = 2$, when varying the tile size, for multiple images. Results are relative to those of the largest tile.

## 4. Experimental Results

In this section we describe the setting of our experiments and analyze our results. We provide several comparisons testing the performance of our compression scheme using our proposed transforms, in both progressive lossy-to-lossless and lossless settings. We should note that regarding the rate computations, we compute the average of the entropies of each compressed component instead of using any particular entropy encoder to discard any bias introduced by an entropy encoder fine-tuned for a specific transform. All experiments have been performed using a prototype encoder implementation in Matlab R2017b. Initially we experiment solely on spatial transforms. We compare our proposed transforms against the GraphBior transform, with and without dividing the components into tiles as well as the reversible 5/3 integer wavelet transform (DWT). All GraphBior transforms are using the maximally flat GraphBior(1,1) filters. The spectral graph lifting designs are using the maximum flatness dual (with the $B_{7,3}$ Parametric-Bernstein-Polynomial for ISGL$_D$) and quad (with the $B_{3,1}$ Parametric-Bernstein-Polynomial for ISGL$_Q$) kernel designs without normalization parameters [41]. We provide further comparisons between the spatial transforms by including spectral transforms such as the Reversible Karhunen Loeve Transform (RKLT) [42] and the DWT.

We have repeated our experiments on crops of the Aviris Yellowstone scene 00, the Hyperion images Lake Monona and Mt. St. Helens and the Landsat Agriculture image. The dimensions of the cropped images that have been used are displayed in Table 1. More precisely, we have discarded the last rows and columns of each component for both Aviris as well as the Landsat scenes. Regarding the Hyperion images we have discarded the last columns and the first 1350 and 760 rows for Lake Monona and Mt. St. Helens respectively.

Next we explore the version of the compression scheme where we include a spectral transform on each band group before applying the spatial transform. The graph transforms are always computed from the luminance values of components and not the spectrally transformed results. We experiment with the Reversible Karhunen Loeve Transform (RKLT) [42] and DWT as spectral transforms. The spatial transforms compared in this experiment are the proposed IGB, the tiled GraphBior, the DWT, the ISGL$_Q$ and ISGL$_D$. In this final experiment, we also include a comparison of our overall compression scheme with the CCSDS-123.0-B-2 standard, tuned according to Reference [19].

### 4.1. Parameter Variations

We first study the effect that different values of $\omega$ have on IGB, without taking into account spectral transforms. To search for the most beneficial choices of $\omega$ to IGB we use tiles of large size.

In Figure 6 the relative rate-distortion plot displays the decrease of performance when larger values of $\omega$ are compared against the smallest $\omega = 2$. One can clearly conclude that it is beneficial to choose smaller values of the parameter $\omega$ when no spectral transform is used. We repeat the experiment with several other hyperspectral images and display the results in Figure 7. Specifically, it is for $\omega = 2$ that tends to lead to optimal results in all cases. This result agrees with our expectations since components closer in the spectral dimension tend to have high correlation. Thus, the component that has been used to create the IGB transform will capture sometimes more accurately the spatial redundancies of its neighboring component rather than of one located further away in the spectral dimension. Though this can lead to detrimental results for low $\omega$ values, when high quantization occurs. As we can see from Figures 6 and 7c,d at low rates, larger parameters $\omega$ outperform lower ones.

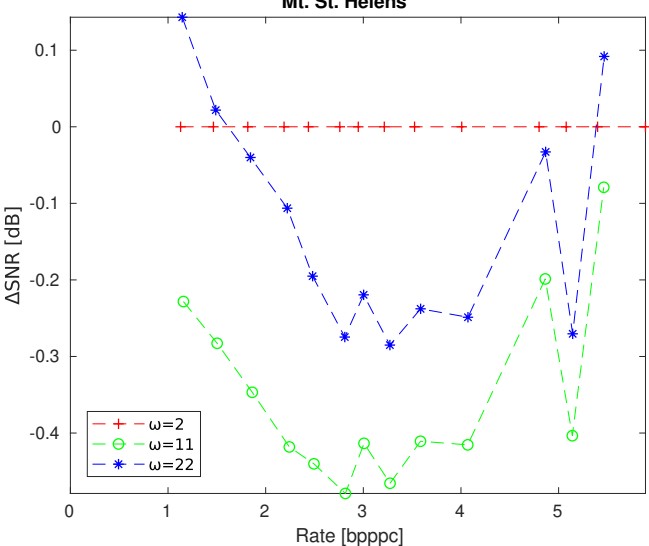

**Figure 6.** Relative rate-distortion plot for IGB using tiles of 16 by 16 when varying $\omega$. The reference curve coincides with tuning the parameter $\omega = 2$.

This could be explained by high quantization, when the surviving features preserved in a decompressed component are general enough to construct a graph that represents sufficiently well a larger number of components. Furthermore, the smaller the parameter $\omega$, the higher the probability of outlier components (ones that do not share high correlation with neighboring components such

as dead bands) to be used for the graph creation, which may result in poor reconstruction results of components of an entire band group.

From Table 3 we also observe that the entropy where the compression scheme using the IGB transform becomes lossless, increases with $\omega$. This explains why at high rates, higher $\omega$ values can be seen to sometimes perform better. This is also noticeable in classical reversible integer wavelet transforms. Their rate distortion curves tend to plateau close to the bitrate at which they achieve a reversible compression. As a result, sometimes, they are surpassed by the distortion curves corresponding to methods that become reversible at higher bitrates (since they still continue to grow with a higher gradient).

**Table 3.** Rates at which IGB achieves a lossless compression for different values of $\omega$. The tiles size is set to 16 by 16. The spectral size of the hyperspectral image should be a multiple of $\omega$, resulting in empty cells. Units are in bpppc.

| Image | IGB | | | | | | |
|---|---|---|---|---|---|---|---|
| | $\omega = 2$ | $\omega = 3$ | $\omega = 8$ | $\omega = 11$ | $\omega = 14$ | $\omega = 16$ | $\omega = 22$ |
| Yellowstone sc. 0 cal. | 6.95 | | 7.00 | | 7.06 | 7.14 | |
| Yellowstone sc. 0 raw | 9.02 | | 9.06 | | 9.14 | 9.15 | |
| Lake Monona | 6.31 | | | 6.36 | | | 6.37 |
| Mt. St. Helens | 6.68 | | | 6.75 | | | 6.75 |
| Agriculture | 4.27 | 4.39 | | | | | |

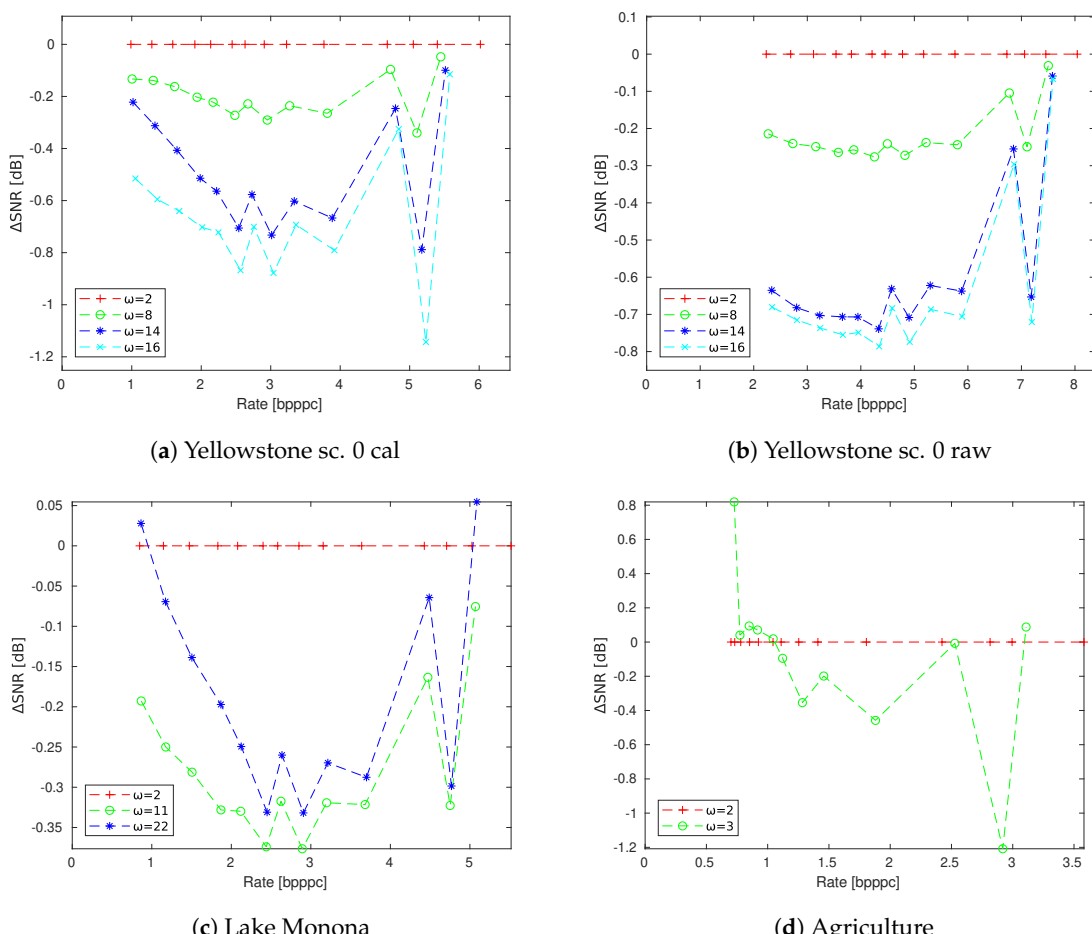

(**a**) Yellowstone sc. 0 cal

(**b**) Yellowstone sc. 0 raw

(**c**) Lake Monona

(**d**) Agriculture

**Figure 7.** Relative rate-distortion plots for IGB using tiles of 16 by 16 when varying the parameter $\omega$ for multiple images. Results are relative to those of smallest value of $\omega$.

### *4.2. Spatial Transformations*

Adopting our previous findings regarding the parameter $\omega$ as well as the size of the tiles, we compare our proposed reversible transforms against the DWT and the GraphBior with and without using tiles. All transforms are applied spatially whereas spectral transforms are not included in the subsequent experiments. It should be noted that all spatial transforms are tested using the compression scheme introduced in Section 3.

For this experiment, the rate-distortion plot, as well as the relative rate-distortion plot using the DWT transform as reference can be observed in Figure 8a,b respectively. From the rate-distortion plots of Figure 8 we observe that the ISGL$_Q$ performs the best for low and medium rates, whereas GraphBior provides the best results for high rates. It is important to mention, though, that in most cases IGB slightly outperforms the DWT as well as the ISGL$_D$ at low rates. Moreover, although in general, tiled GraphBior outperforms IGB, at low rates, both perform similarly. This same behavior is also observed between classical discrete wavelet transforms and their reversible counterparts. Given the previous observation and since the results of IGB improve as the tile size increases, one could speculate that a IGB that does not use any tiles would perform similarly to GraphBior for low rates. This experiment has been repeated on different hyperspectral scenes and the results can be observed in Figure 9.

In Table 4 we can see the rates at which the IGB, DWT, ISGL$_Q$ and ISGL$_D$ achieve a lossless compression. We also observe that in most cases the entropy where IGB achieves a lossless compression is the lowest.

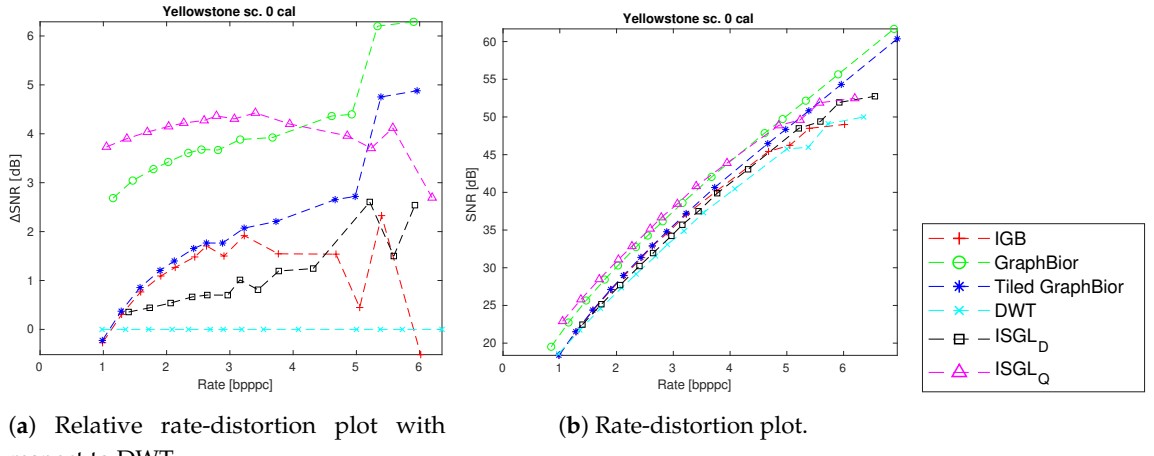

(**a**) Relative rate-distortion plot with respect to DWT.

(**b**) Rate-distortion plot.

**Figure 8.** Comparison of spatial transforms. Tiles of 16 by 16 and $\omega = 2$ were used.

**Table 4.** Rates at which each integer transform achieves a lossless compression. No spectral transform is used. The parameter $\omega$ is set to 2. The IGB transform uses tiles of 16 by 16 for the Yellowstone images and 32 by 32 for Lake Monona, Mt. St. Helens and Agriculture. Units are in bpppc.

| Image | Transform | | | |
|---|---|---|---|---|
| | IGB | DWT | ISGL$_D$ | ISGL$_Q$ |
| Yellowstone sc. 0 cal. | 6.95 | 7.29 | 7.44 | 7.11 |
| Yellowstone sc. 0 raw | 9.02 | 9.32 | 9.47 | 9.19 |
| Lake Monona | 6.26 | 6.23 | 6.50 | 6.28 |
| Mt. St. Helens | 6.63 | 6.57 | 6.93 | 6.67 |
| Agriculture | 4.21 | 4.37 | 4.67 | 4.37 |

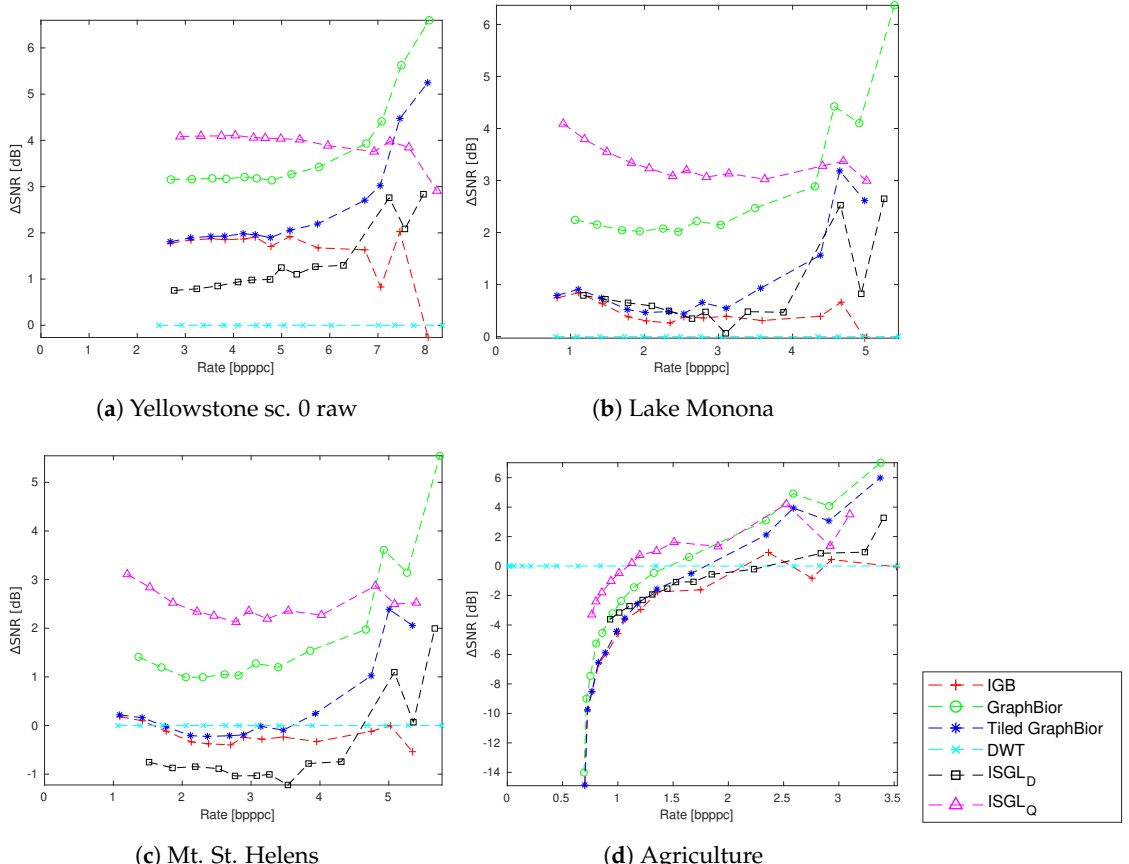

(**a**) Yellowstone sc. 0 raw

(**b**) Lake Monona

(**c**) Mt. St. Helens

(**d**) Agriculture

**Figure 9.** Relative rate-distortion plots comparing spatial transforms using $\omega = 2$ for multiple images. Results are relative to DWT. The tiles sizes are 16 by 16 for the Yellowstone sc. 0 raw image and 32 by 32 for the rest.

### 4.3. Spectral and Spatial Transformations

Our comparisons on strictly spatial transforms are ensued by experiments that include the DWT and the RKLT as spectral transforms, in the compression scheme mentioned in Section 3. Succeeding the spectral transform, we then compare our proposed reversible transforms against the DWT and the tiled GraphBior applied in the spatial dimension of the hyperspectral images. Our overall compression scheme including classical spectral and graph spatial transforms is also compared against the CCSDS-123.0-B-2 standard.

In Figure 10a,b we present the rate-distortion plot as well as the relative rate-distortion plot. The results are relative to applying a spectral RKLT along the entire hyperspectral image, followed by a spatial DWT.

Regarding the comparison between the transforms used in our compression scheme, we observe that a spectral RKLT followed by the proposed ISGL$_Q$ or ISGL$_D$ transforms perform the best. Several additional comparisons are done on multiple hyperspectral images and from our results in Figure 11 we observe that the spectral RKLT and the spatial ISGL$_Q$ systematically outperforms the spectral RKLT and spatial DWT, mostly at medium to high rates. The spectral RKLT and spatial ISGL$_D$, also, usually surpasses the reference method but provides less important results when compared to using the spatial ISGL$_Q$ instead. On the other hand the spectral RKLT and the spatial IGB is almost always providing worse results when compared to the spectral RKLT and the spatial DWT.

Our conclusions from the comparison of the spatial transforms done in Section 4.2 are also evident in our current experiment. Regardless of the spectral transform, the spatial IGB coincides with the spatial tiled GraphBior for low entropy values. When we apply DWT as a spectral transform, out of the spatial transforms, the ISGL$_Q$ performs the best followed mostly by the DWT and the ISGL$_D$.

From Table 5, we observe that mostly the spectral RKLT and the spatial DWT achieve a lossless compression at lower rates. Only in the case of the Hyperion images (Lake Monona and Mt. St. Helens) the spectral RKLT followed by the spatial ISGL$_Q$ performs better.

When comparing the compression efficiency of the latest CCSDS-123.0-B-2 standard against our overall proposed compression scheme, we observe the usual pattern where transform-based methods tend to perform better at lower rates while the CCSDS standard yields better results at mid to high rates. In Figures 10 and 11, a spectral RKLT followed by a spatial DWT often outperforms CCSDS for low rates. Furthermore, CCSDS results are also improved by the proposed compression scheme using a spectral RKLT and a spatial ISGL$_Q$ for low-rate results in Figure 11a,b. In the mid to high rate regions, the CCSDS standard clearly outperforms all transform-based methods. In addition, for pure lossless compression, the CCSDS standard consistently achieves best results (Table 5). It should be noted though that the CCSDS standard is a highly refined method, whereas the proposed compression scheme here presented is mainly designed to compare the graph transforms. Thus, it lacks many of the techniques that the CCSDS standard incorporates, such as quantization enhancements, noise tolerance or entropy encoding with adaptive statistics.

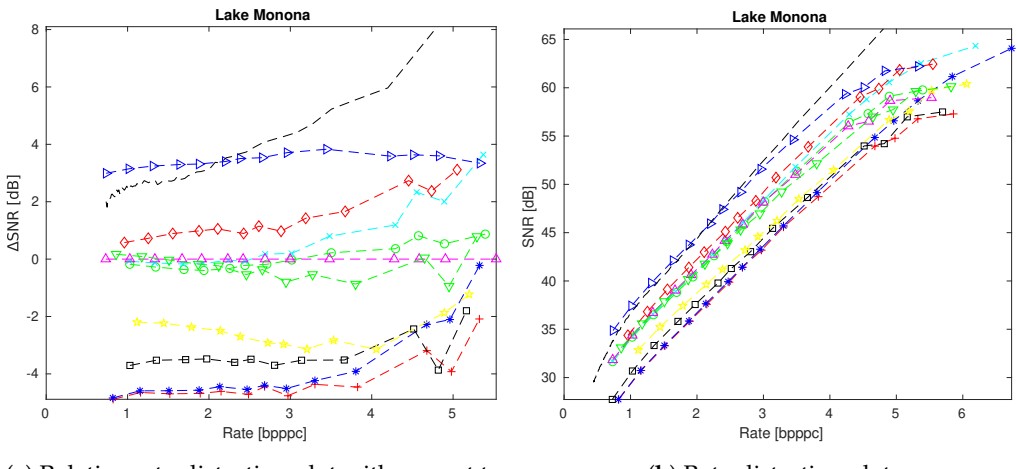

(**a**) Relative rate-distortion plot with respect to the spectral RKLT followed by the spatial DWT.

(**b**) Rate-distortion plot

| | |
|---|---|
| — + — | DWT + IGB |
| — ⊖ — | RKLT + IGB |
| — ✳ — | DWT + Tiled GraphBior |
| — ✕ — | RKLT + Tiled GraphBior |
| — ⊟ — | DWT + DWT |
| — △ — | RKLT + DWT |
| — ☆ — | DWT + ISGL$_D$ |
| — ◇ — | RKLT + ISGL$_D$ |
| — ▽ — | DWT + ISGL$_Q$ |
| — ▷ — | RKLT + ISGL$_Q$ |
| — — — | CCSDS |

**Figure 10.** Comparison of spectral + spatial transforms. Tiles of 16 by 16 and $\omega = 11$ was used.

**Table 5.** Rates at which each of the integer transforms achieves a lossless compression. RKLT and DWT are used as spectral transforms. For the methods that use tiles, their size has been set to 16 by 16. The Yellowstone images are evaluated at $\omega = 8$, the Lake Monona and Mt. St. Helens are evaluated at $\omega = 11$ and the Agriculture image is evaluated at $\omega = 3$. For each column, the spectral transform is mentioned first followed by the spatial transform (spectral transform + spatial transform). Units are in bpppc.

| Image | Transforms | | | | | | | | CCSDS |
|---|---|---|---|---|---|---|---|---|---|
| | DWT + IGB | RKLT + IGB | DWT + DWT | DWT + ISGL$_D$ | DWT + ISGL$_Q$ | RKLT + DWT | RKLT + ISGL$_D$ | RKLT + ISGL$_Q$ | |
| Yellowstone sc. 0 cal. | 5.03 | 4.41 | 4.73 | 5.36 | 5.05 | 3.74 | 4.65 | 4.36 | 4.04 |
| Yellowstone sc. 0 raw | 7.17 | 6.47 | 6.97 | 7.54 | 7.25 | 5.93 | 6.72 | 6.44 | 6.19 |
| Lake Monona | 6.69 | 6.19 | 6.56 | 6.89 | 6.66 | 6.35 | 6.34 | 6.13 | 6.10 |
| Mt. St. Helens | 7.06 | 6.52 | 6.90 | 7.32 | 7.05 | 6.58 | 6.71 | 6.47 | 6.37 |
| Agriculture | 4.21 | 3.96 | 3.96 | 4.63 | 4.34 | 3.68 | 4.37 | 4.08 | 3.62 |

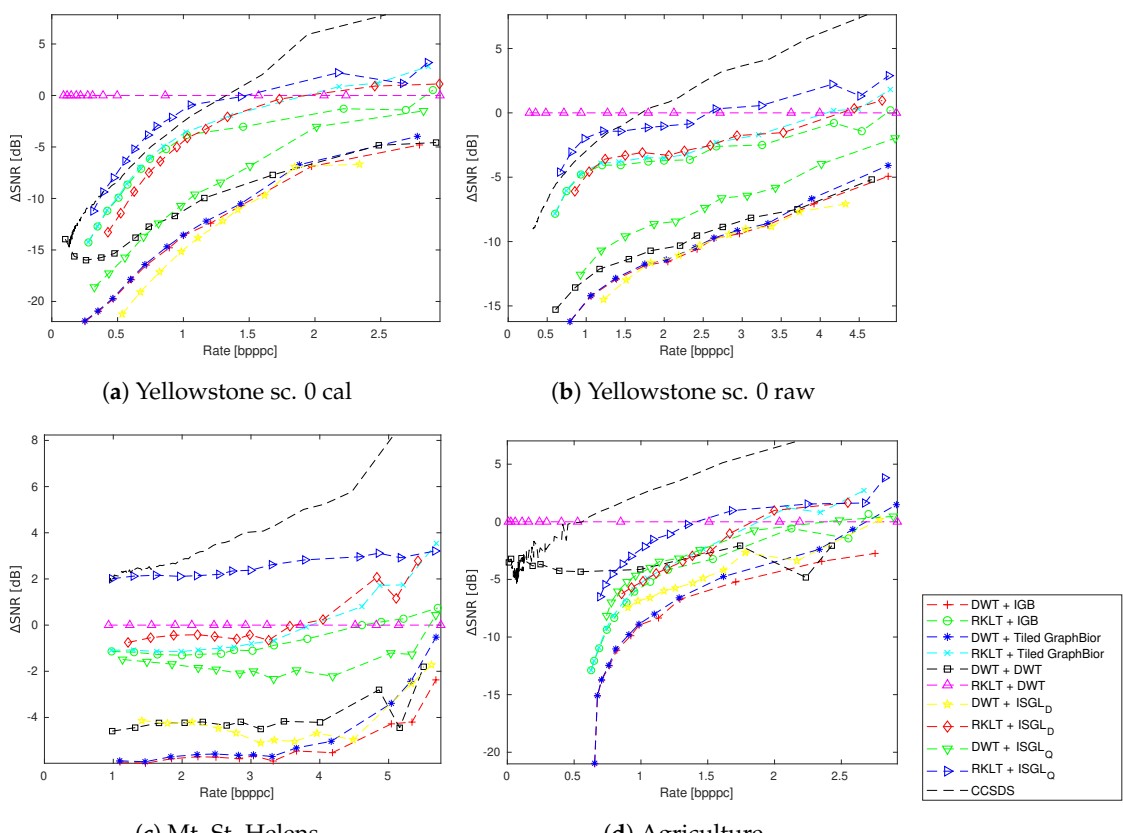

(**a**) Yellowstone sc. 0 cal

(**b**) Yellowstone sc. 0 raw

(**c**) Mt. St. Helens

(**d**) Agriculture

**Figure 11.** Relative rate-distortion plots comparing spectral + spatial transforms for multiple images. Results are relative to the spectral RKLT followed by the spatial DWT. Tiles of 16 by 16 were used. The parameter $\omega$ is set to 8 for Yellowstone sc. 0 cal., to 11 for Mt. St. Helens and to 3 for Agriculture.

## 5. Conclusions

In this paper we study the suitability of graph transforms for lossy-to-lossless hyperspectral compression. The adopted compression scheme organizes the components into packets, called band groups and transforms each one of them through graph wavelet filterbanks. We introduce two spatial, integer-to-integer, biorthogonal graph filterbank transforms. The first transform is calculated by applying a TERM factorization on the GraphBior filterbank, whereas the second one is designed by modifying the spectral graph lifting transform [34]. The high computational complexity of the TERM decomposition is addressed and is solved through processing the GraphBior transform in tiles. Our theoretical interpretations as well as our experimental results show that it is advantageous to use larger tiles and give us valuable insight about the choice of sizes of band groups. Our experiments without spectral transformations suggest that it is preferable to use small band groups, while our

integer-to-integer transforms outperform the DWT in the lossy regime. Further-more we show that one of our integer transforms performs similarly to the DWT for the case of lossless compression. Additional experimental results including spectral transforms on each bandgroup show that our proposition improves on the results obtained by using the spectral RKLT along all the spectral dimension of the hyperspectral image followed by the spatial DWT in the lossy setting, as well as, in some cases, at the lossless one.

**Author Contributions:** Conceptualization, D.E.O.T., K.C., I.B. and J.S.-S.; methodology, D.E.O.T., K.C., I.B. and J.S.-S.; software, D.E.O.T.; validation, D.E.O.T., I.B. and J.S.-S.; formal analysis, D.E.O.T., K.C., I.B. and J.S.-S. ; investigation, D.E.O.T., K.C., I.B. and J.S.-S.; resources, D.E.O.T., K.C., I.B. and J.S.-S.; data curation, D.E.O.T., I.B. and J.S.-S.; writing–original draft preparation, D.E.O.T.; writing–review and editing, D.E.O.T., I.B. and J.S.-S.; visualization, D.E.O.T., I.B. and J.S.-S.; supervision, I.B. and J.S.-S.; project administration, I.B. and J.S.-S.; funding acquisition, I.B. and J.S.-S.

**Funding:** This research was funded by the Spanish Ministry of Economy and Competitiveness and the European Regional Development Fund under grants RTI2018-095287-B-I00 and TIN2015-71126-R (MINECO/FEDER, UE) and BES-2016-078369 (Programa Formación de Personal Investigador), and by the Catalan Government under grant 2017SGR-463.

**Conflicts of Interest:** The authors declare no conflict of interest.

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
