# Peer review of "Compression of Hyperspectral Scenes through Integer-to-Integer Spectral Graph Transforms†"

_remotesensing, doi:10.3390/rs11192290_

Round 1

Reviewer 1 Report

Technical Comments

a- Line 72: Perhaps there is a typo,  the text says,

“we provide transforms that allow us to construct a lossless-to-lossy compression scheme for hyperspectral images”

   But, the objective of the paper is to introduce a lossless compression technique.

On the other hand in line 85 you state

“Using our proposed transforms mentioned above, we design a lossy-to-lossless extension”

b- The authors make an exhaustive analysis of the performance of their algorithm and compare it with other methods. Perhaps, they could add  brief parragraf or section on the advantatges of their method related to the storage of multiespectral images, memory savings, time of decoding, etc.

c- The summary in section 3.1, assumes that the reader is familiar with reference [22]. The relationship between the construction of the graph from components and the generation of both bipartite graphs is not clear from the summary.

Reference format:

*Check Reference 23

English:

*Line 21, Said transforms -> do you mean “Above mentioned transforms”?

*Line 202 it is highly time consuming to factorize -> perhaps you mean “the factorization is computationally  intensive”

*Line361: sentence ‘Our experiments devoid’ is not clear.

Reviewer 2 Report

The paper presents a method for the compression of hyperspectral scenes. The proposed method which is loss less adopts an integer-to-integer transforms which is also used for biorthogonal graph filterbanks. 

Minor Comments:

1)Please check the grammar and typos. For example, some of the minor typos are:

- There is duplicate "the" used in abstract line 6. 

- Page2, line 64, check the spell of "filterbank".

-Page 5,line 179, check the spell of "biorthogonal ".

-Page 6,line 201, there is duplicate "the" used in the sentence.

Please check for other typos and minor English grammar check as well.

Reviewer 3 Report

This is a paper written in a very comprehensive way in perfect English language. The compression of hyperspectral images through integer-to-integer spectral graph transforms. Several parameters vary to find the ideal parameters for lossless compression. However there are three major issues on my opinion:

1) No comparison is given with other lossy and lossless compression techniques in terms of compression efficiency, speed, etc

2) There is no presentation of the overall theoretical modelling of the system. Instead, there are references to the transforms used. This gives me the impression of the authors combined some off the shelf algorithms and tuned them without any further development and integration performed.

3) There is absolutely no information about the environment where the algorithms and the transforms have been tested

Round 2

Reviewer 3 Report

Concerning my comments the authors

a) have clarified that all the tests have been performed in MATLAB

b) compared their achievements with CCSDS as a reference

c) have not presented any speed report in order to avoid unfair comparison

Although clarifications have been given about the construction of bipartite graphs an overall theoretical description of the proposed method that would incorporate the transforms and other methods used is still missing.

The aim of this paper is to explore the potential of graph transforms and the authors focused on making the overall scheme feasible, while selecting a very simple quantizer. According to the authors they have not exploited any further statistical redundancy in the entropy encoding stage. Moreover, the authors confirm that they have not integrated their method in a new tool. Although the results are interesting I still insist that the contribution of this paper is not adequate for a high-impact factor journal.